

# Gender-specific linkages: frailty, polypharmacy, anti-cholinergic burden, and 5-year mortality risk—a real-world analysis

Yin Yi Chou[1,2], Yu Shan Lee[1,3], Chu Sheng Lin[4,5], Jun Peng Chen[6], Fu-Hsuan Kuo[1,3], Cheng-Fu Lin[1,5,7], Yi-Ming Chen[2,4,6,8,9] and Shih-Yi Lin[1,4,10]

[1] Center for Geriatrics & Gerontology, Taichung Veterans General Hospital, Taichung, Taiwan
[2] Division of Allergy, Immunology and Rheumatology, Taichung Veterans General Hospital, Taichung, Taiwan
[3] Division of Neurology, Taichung Veterans General Hospital, Taichung, Taiwan
[4] Department of Post-Baccalaureate Medicine, College of Medicine, National Chung Hsing University, Taichung, Taiwan
[5] Department of Family Medicine, Taichung Veterans General Hospital, Taichung, Taiwan
[6] Department of Medical Research, Taichung Veterans General Hospital, Taichung, Taiwan
[7] Division of Occupational Medicine, Department of Emergency, Taichung Veterans General Hospital, Taichung, Taiwan
[8] Institute of Biomedical Science and Rong-Hsing Research Center for Translational Medicine, Chung Hsing University, Taichung, Taiwan
[9] School of Medicine, National Yang Ming Chiao Tung University, Taipei, Taiwan
[10] Division of Endocrinology and Metabolism, Department of Internal Medicine, Taichung Veterans General Hospital, Taichung, Taiwan

Corresponding authors
Yi-Ming Chen,
ymchen1@vghtc.gov.tw
Shih-Yi Lin, sylin@vghtc.gov.tw

## ABSTRACT

**Background**. With higher age, frailty escalates the risk of falls, unexpected physical dysfunction, hospitalization, and mortality. Polypharmacy in the older population is a major challenge that not only increases medical costs, but also may worsen the risk of hospitalization and death. More importantly, the properties of anti-cholinergic drugs contribute various negative effects. This study aimed to investigate the sex difference in the association of polypharmacy, anticholinergic burden, and frailty with mortality.

**Methods**. Participants older than 65 years who attended the geriatric outpatient clinic of the study center between January 2015 and July 2020 were invited to participate in this retrospective study. Comprehensive geriatric assessment data were collected and the phenotype of frailty was determined by Fried's criteria. Cox regression and the Kaplan–Meier curve were used to identify risk factors of 5-year survival along with intergroup differences in the risks.

**Results**. Of the 2,077 participants, 47.5% were female. The prevalence of frailty and the rate of polypharmacy were 44.7% and 60.6%, respectively. Higher age, male sex, low body mass index, low Mini-Mental State Examination scores, low activities of daily living, frailty status, polypharmacy, and a high Charlson Comorbidity Index score, and greater anticholinergic burden were significant risk factors that were associated with the 5-year all-cause mortality. Male patients with frailty exhibited the highest risks of mortality compared with male patients without frailty and female patients with or without frailty. Polypharmacy was significantly associated with a higher 5-year mortality rate in the frail male group compared with the non-frail male. In frail female group,

individuals with a higher anticholinergic burden (as indicated by the Anticholinergic Cognitive Burden Scale) from drug usage exhibited an elevated 5-year mortality rate. **Conclusions**. Polypharmacy and greater anticholinergic burden, synergistically interacted with frailty and intensified the 5-year mortality risk in a gender-specific manner. To mitigate mortality risks, clinicians should prudently identify polypharmacy and anticholinergic burden in the older population.

## INTRODUCTION

The World Health Organization forecasted that the global population of individuals older than 60 years will nearly double, from 12% to 22%, between 2015 and 2050 (*Officer et al., 2020*). With the rapid growth rate of the older population, healthcare systems are facing a surging demand and hitherto unexperienced challenges. Older adults are more vulnerable, have more comorbidities, and have a higher risk of hospitalization and readmission that confers a huge medical burden and may even lead to ineffective medical treatment.

Older adults with frailty tend to be vulnerable to stress (*Xue, 2011*) and have an increased risk of falls, physical disability, hospitalization, and mortality (*Clegg et al., 2013*; *Ensrud et al., 2009*; *Xue, 2011*). The prevalence of frailty increases with higher age and in females (*O'Caoimh et al., 2021*; *Yang et al., 2018*). Although frailty is more prevalent in the female population, mortality is higher in the male population (*Collard et al., 2012*; *Gordon et al., 2017*). Thus, women seem to be more tolerant of frailty-associated adverse outcomes. The sex differences in frailty and mortality among older individuals constitute the so-called the sex-frailty paradox. However, no clear explanation or conclusive evidence for this paradox has been reported so far (*Gordon & Hubbard, 2020*; *Gordon et al., 2017*).

In recent years, some studies have predicted the long-term mortality of older outpatients and community-dwelling older adults (*Cheong et al., 2021*; *Graham et al., 2019*; *Guerreiro et al., 2022*; *Rhodius-Meester et al., 2021*). The Amsterdam Ageing Cohort classified the common geriatric syndrome into five domains. Frailty was categorized in the physical domain of the aforementioned study. The results showed that the five geriatric domains of are not only highly prevalent, but also cumulatively associated with mortality (*Rhodius-Meester et al., 2021*). However, sex differences in the association of mortality and geriatric syndrome remain unknown.

In addition to frailty, several aggravating factors are related to adverse outcomes in the geriatric population. Frailty in combination with delirium (*Dani et al., 2018*), dementia (*Maxwell et al., 2019*), and depression (*Soysal et al., 2017*) could exacerbate the mortality risk than just frailty alone. Polypharmacy increases the risks of hospitalization and mortality (*Chang et al., 2020*). Especially in cases of polypharmacy, it is important to recognize that even medications with minimal anticholinergic effects can have a significant impact on creating an anticholinergic burden (*Sumbul-Sekerci et al., 2022*). Elderly women could exhibit a higher vulnerability to drug-related adverse effects compared to elderly men,

partially attributed to alterations in pharmacokinetics and pharmacodynamics (*Soldin & Mattison, 2009*). The effect of polypharmacy on mortality is more prominent in the pre-frail groups than in the frail group (*Chang et al., 2020*). However, it is unclear whether a sex difference in the mortality risk exists in patients with polypharmacy and frailty. Sex differences in anticholinergic burden have also been observed (*Trenaman et al., 2021*).

This study aimed to investigate whether the sex difference Influences the mortality risk of older people with polypharmacy, anticholinergic burden and frailty in the outpatient clinic setting.

## MATERIALS AND METHODS

### Study design and participants

A single-center, retrospective cohort study was conducted at Taichung Veterans General Hospital, Taiwan. Patients older than 65 years who visited the comprehensive geriatric outpatient clinic from January 2015 to July 2020 were enrolled. Patients without complete data on the Comprehensive Geriatric Assessment (CGA) and frailty status were excluded from the analysis. This study was approved by the Ethics Committee of Clinical Research, Taichung Veterans General Hospital. As the patient data were anonymized prior to analysis, the requirement for written informed consent from patients was waived for this study, which was approved by the same Ethics Committee (CE21219A). Portions of this text were previously published as part of a preprint (https://www.researchsquare.com/article/rs-1889661/v1).

### Data collection

The primary outcome of this study was the 5-year mortality rate, which was ascertained from the data provided by the Taiwan Ministry of Health and Welfare. Frailty was defined using the 5-item Fried phenotype, or Cardiovascular Health Study (CHS) scale and included: exhaustion, unintentional weight loss, weakness, slowness, and low physical activity. Participants with one or two component(s) were defined as pre-frailty. Patients who fulfilled three or more components were classified as frail. Weight loss was determined by an unintentional reduction of more than 5 kg body weight in the past year. Low handgrip strength using cutoffs proposed by the consensus report of the Asian Working Group for Sarcopenia was used to define weakness (*Chen et al., 2014*). Participants were classified as having slowness if the walking speed less than 0.8 m/s. Low physical activity was measured using the weekly average metabolic equivalent of task-hour less than 2.5 in women or 3.75 in men (*Huang et al., 2020*).

Comorbidities are based on the patient's self-reported chronic disease and included: diabetes mellitus (DM), hypertension (HTN), cerebrovascular accident (CVA), coronary artery disease (CAD), chronic kidney disease (CKD), and chronic obstructive pulmonary disease (COPD). The Charlson's Comorbidity Index (CCI) was used to measure comorbid conditions (*Charlson et al., 1994*). During the CGA assessment, the case manager would review the patient's medication regimen, and if the number of medications exceeds five, the patient would be classified as having polypharmacy (*Rankin et al., 2018*). The anticholinergic burden was assessed using the anticholinergic cognitive burden (ACB)

score, assigning a score of 1, 2, or 3 to each medication based on its anticholinergic potency. A medication with a score of 1 exhibits mild anticholinergic effects, whereas a score of 3 signifies strong anticholinergic effects (*Rudd et al., 2005*). We employed the electronic medical record system to capture participants' medication records for the 6-month period both prior to and following the CGA evaluation date.

## Comprehensive geriatric assessment

All participants underwent CGA that was undertaken by trained case managers. The patients' basic demographics, including age, sex, and BMI were recorded. Cognitive function was evaluated by the Chinese version of Mini-Mental State Examination (MMSE) (*Folstein, Folstein & McHugh, 1975*). The Geriatric Depression Scale-5 item (GDS-5) was used to screen for depressive symptoms (*Hoyl et al., 1999*). Physical function was assessed by activities of daily living (ADL) (*Mahoney & Barthel, 1965*).

## Statistical analysis

The chi-square test or the Mann–Whitney $U$ test were used to compare variables between participants in the survival and mortality groups. Cox regression analysis was performed to investigate the risk factors that were independently associated with mortality after adjusting for age, sex, BMI, MMSE, GDS-5, ADL, frailty status polypharmacy, and CCI, for variables with a $p$-value less than 0.05 in the univariable analysis. We used Kaplan–Meier survival analysis to determine the sex differences in 5-year survival by the presence of frailty and polypharmacy. Pairwise comparisons of survival curves were analyzed by the log-rank test. Statistical Package for the Social Science (SPSS) version 22.0 (IBM Corp., Armonk, NY, USA) was used to analyze all of the data. Significance was set at $p < 0.05$.

# RESULTS

## Demographics, CGA, and comorbidity of participants

From January 2015 to July 2020, a total of 2,077 patients completed the CGA and the assessment of frailty. At the 5-year follow-up, 304 participants had died (14.6%). We compared the basic demographics and CGA parameters between the survival and mortality groups (Table 1). Compared with the survival group, the mortality group had lower BMI (24.4 kg/m² *vs.* 23.2 kg/m, $p < 0.001$), lower MMSE scores (26 *vs.* 22.5, $p < 0.001$), poorer ADL (100 *vs.* 90, $p < 0.001$), and a significantly higher proportion of frailty (40.9% *vs.* 66.8%, $p < 0.001$). In addition, the mortality group had higher prevalence of CVA (19.7% *vs.* 11.4%, $p < 0.001$), CAD (37.5% *vs.* 29.9%, $p < 0.05$), CKD (38.2% *vs.* 29.2%, $p < 0.05$), dementia (28.3% *vs.* 20.8%, $p < 0.05$), COPD (18.4% *vs.* 7.5%, $p < 0.001$) polypharmacy (69.4% *vs.* 57.8%, $p < 0.05$), and greater ACB score (60.3% *vs.* 77.0%, $p < 0.001$) compared with the survival group.

## Risk factors for mortality

Table 2 presents the risk factors that were significantly associated with all-cause mortality. We discovered that higher age (HR: 1.07, 95% CI [1.05–1.09]), male sex (HR: 1.86, 95% CI [1.33–2.61]), low BMI (HR: 0.94, 95% CI [0.90–0.98]), low MMSE (HR: 0.96, 95%

**Table 1  Baseline characteristics of the participants.**

| | Survival ($n = 1,773$) | | Mortality ($n = 304$) | | $p$-value |
|---|---|---|---|---|---|
| Age, years | 77.0 | (71.0–84.0) | 85.0 | (80.0–89.0) | <0.001[**] |
| Sex | | | | | <0.001[**] |
|    Male | 874 | (49.3%) | 217 | (71.4%) | |
|    Female | 899 | (50.7%) | 87 | (28.6%) | |
| BMI | 24.4 | (22.1–26.9) | 23.2 | (20.9–25.5) | <0.001[**] |
| MMSE | 26.0 | (21.0–28.0) | 22.5 | (16.8–27.0) | <0.001[**] |
| GDS-5 | 1.0 | (0.0–2.0) | 1.0 | (0.0–2.0) | <0.001[**] |
| ADL | 100.0 | (90.0–100.0) | 90.0 | (71.3–100.0) | <0.001[**] |
| Frailty status | | | | | |
|    Non-frailty | 1,047 | (59.1%) | 101 | (33.2%) | <0.001[**] |
|    Frailty | 726 | (40.9%) | 203 | (66.8%) | |
| DM | 920 | (51.9%) | 137 | (45.1%) | 0.033[*] |
| HTN | 1,216 | (68.6%) | 208 | (68.4%) | 1.000 |
| CVA | 203 | (11.4%) | 60 | (19.7%) | <0.001[**] |
| CAD | 530 | (29.9%) | 114 | (37.5%) | 0.010[*] |
| CKD | 517 | (29.2%) | 116 | (38.2%) | 0.002[**] |
| Dementia | 369 | (20.8%) | 86 | (28.3%) | 0.005[**] |
| Parkinson | 101 | (5.7%) | 24 | (7.9%) | 0.174 |
| COPD | 133 | (7.5%) | 56 | (18.4%) | <0.001[**] |
| Polypharmacy | 1,024 | (57.8%) | 205 | (67.4%) | 0.002[**] |
| CCI | 2 | (1–3) | 2 | (1–3) | 0.994 |
| ACB score | | | | | <0.001[**] |
|    ACB score 0 | 703 | (39.7%) | 70 | (23.0%) | |
|    ACB score 1–2 | 437 | (24.6%) | 77 | (25.3%) | |
|    ACB score 3+ | 633 | (35.7%) | 157 | (51.6%) | |

**Notes.**
Data are presented as number (percentage) or median (interquartile range).
[*]$p < 0.05$.
[**]$p < 0.005$ obtained by the chi-square test or the Mann–Whitney $U$ test.
Abbreviation: BMI, body mass index; MMSE, Mini-Mental State Examination; GDS-5, Geriatric Depression cale-5 item; ADL, activities of daily living; CHS, Cardiovascular Health Study; DM, diabetes mellitus; HTN, hypertension; CVA, cerebrovascular accident; CAD, coronary artery disease; CKD, chronic kidney disease; COPD, chronic obstructive pulmonary disease; Polypharmacy, taking 5 or more medications daily; CCI, Charlson Comorbidity Index; ACB, Anticholinergic Burden.

CI [0.93–0.99]), ADL (HR: 0.99, 95% CI [0.98–0.99]), high CCI score (HR: 1.22, 95% CI [1.03–1.21]), and high ACB score (HR:1.88, 95% CI [1.30–2.71]) were associated with mortality.

## Five-year survival by sex, frailty, polypharmacy, and anticholinergic burden

To investigate the sex difference of the interaction between frailty and polypharmacy in older patients, the 5-year survival analysis was performed using Kaplan–Meier plots. Figures 1A and 1B shows that the male group exhibited a lower survival rate than the female group, both in the frailty *vs.* non-frailty groups and the polypharmacy *vs.* non-polypharmacy group. Figure 1A shows that the male frailty group had the worst prognosis

**Table 2 Cox regression analysis of the risks of 5-year all-cause mortality.**

| | Univariate | | | Multivariable | | |
|---|---|---|---|---|---|---|
| | Hazard ratio | 95% CI | p value | Hazard ratio | 95% CI | p value |
| Age, years | 1.10 | (1.09–1.12) | <0.001** | 1.07 | (1.05–1.09) | <0.001** |
| Sex | | | | | | |
|   Female | Reference | | | Reference | | |
|   Male | 2.08 | (1.62–2.67) | <0.001** | 1.86 | (1.33–2.61) | <0.001** |
| BMI | 0.91 | (0.88–0.95) | <0.001** | 0.94 | (0.90–0.98) | 0.002** |
| MMSE | 0.92 | (0.90–0.93) | <0.001** | 0.96 | (0.93–0.99) | 0.004** |
| GDS-5 | 1.22 | (1.12–1.32) | <0.001** | 1.11 | (1.01–1.22) | 0.034* |
| ADL | 0.97 | (0.97–0.97) | <0.001** | 0.99 | (0.98–0.99) | 0.001** |
| Frail status | | | | | | |
|   Non-frail | Reference | | | Reference | | |
|   Frail | 2.69 | (2.12–3.41) | <0.001** | 1.19 | (0.88–1.62) | 0.265 |
| Polypharmacy | 1.40 | (1.10–1.78) | 0.006** | 1.21 | (0.87–1.69) | 0.261 |
| CCI | 1.13 | (1.05–1.21) | 0.001** | 1.12 | (1.03–1.21) | 0.005** |
| ACB score | | | | | | |
|   ACB score 0 | Reference | | | Reference | | |
|   ACB score 1–2 | 1.79 | (1.30–2.48) | <0.001** | 1.50 | (1.01–2.23) | 0.047* |
|   ACB score 3+ | 2.57 | (1.94–3.41) | <0.001** | 1.88 | (1.30–2.71) | 0.001** |

**Notes.**

*$p < 0.05$.

**$p < 0.005$ by Cox regression analysis after adjustment for age, sex, BMI, MMSE, GDS-5, ADL, CHS, polypharmacy, and CCI.

Abbreviation: BMI, body mass index; CI, confidence interval; MMSE, Mini-Mental State Examination; GDS-5, Geriatric Depression cale-5 item; ADL, activities of daily living; CHS, Cardiovascular Health Study; Polypharmacy, taking 5 or more medications daily; CCI, Charlson Comorbidity Index; ACB, Anticholinergic Burden.

and the female non-frailty group had the best prognosis (5-year survival rate 56.1% *vs.* 92.8%, log-rank $p < 0.001$). The female frailty group had a similar prognosis as the male non-frailty group (5-year survival rate: 75.6% *vs.* 78.3%, $p = 0.101$). Figure 1B shows that polypharmacy had no effect on mortality in the same sex group. To analyze the sex difference and the association of polypharmacy and frailty with mortality, we categorized the population by the presence of polypharmacy and frailty with regard to different sexes. The all-cause mortality was highest in the frail male group with polypharmacy (Figs. 1C and 1D) compared with the counterparts. However, the effect of polypharmacy on mortality was most pronounced in the male non-frail group.

We categorized the anticholinergic burden into three groups based on ACB scores: 0, 1–2, and greater than or equal to 3. Subsequently, we conducted a detailed analysis of how the level of anticholinergic burden impacts the long-term mortality rates in both male and female groups. The level of anticholinergic burden had no significant effect on long term mortality in the male population except for non-frail males with no anticholinergic exposure (Figs. 2A and 2B). However, the level of anticholinergic drugs has a significant impact on long-term mortality in women, especially non-frail women (Figs. 2C and 2D).

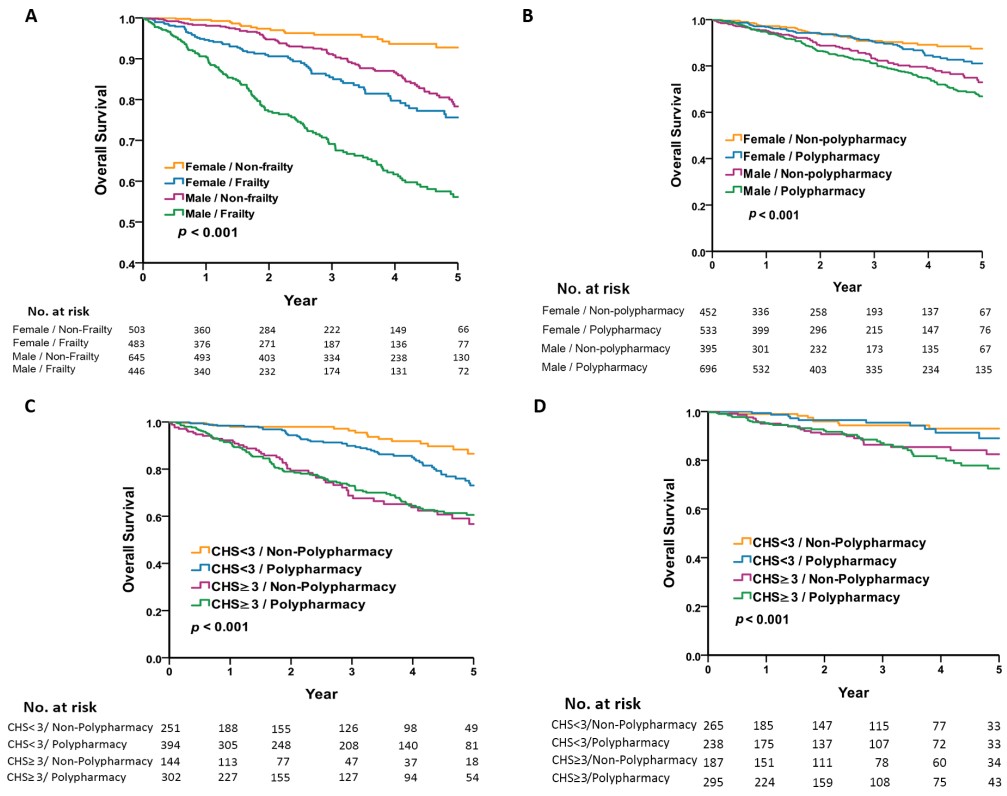

**Figure 1  Five-year survival by sex, frailty, and polypharmacy.** (A) Shows that the male frailty group had the worst prognosis and the female non-frailty group had the best prognosis. (B) Shows that there were significant differences in mortality between males and females, regardless of whether polypharmacy is present or not. (C) Shows that the all-cause mortality was highest in the frail male group with non-polypharmacy. (D) Shows that the all-cause mortality was highest in the frail female group with polypharmacy.

## DISCUSSION

In this study, we found that male sex, lower BMI, lower MMSE, lower ADL, frailty, polypharmacy, higher CCI, and greater anticholinergic burden were risk factors for long-term mortality. We discovered that mortality risks were significantly increased, especially in male patients with frailty. Moreover, polypharmacy was a major prognostic determinant in the male, non-frail group. Furthermore, the anticholinergic burden substantially escalates the mortality rate within the female cohort. Therefore, to improve long-term patient survival, it is prudent to identify and manage polypharmacy and anticholinergic burden.

The prevalence of frailty could vary by assessment tools, region, and different clinical settings. A systematic review study showed that the frailty prevalence in the community setting measured using physical frailty was 12% compared to 24% for the deficit accumulation model (*O'Caoimh et al., 2021*). Using the frailty phenotype as the assessment tool, *Braun et al. (2019)* showed that the frailty prevalence in the outpatient setting was 17.8%. However, another study conducted at Ain Shams University Hospitals, Singapore showed a much higher prevalence: 48%, using the clinical frailty scale as the assessment

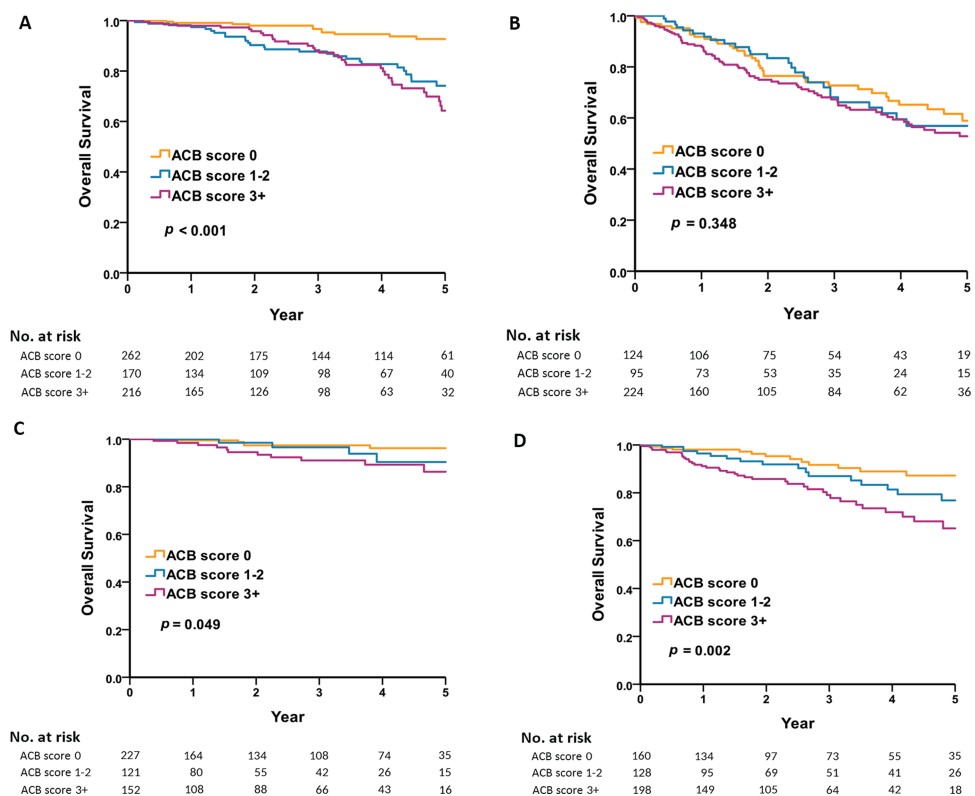

**Figure 2** **Five-year survival by sex, frailty, and anticholinergic burden.** (A) Shows that the level of anticholinergic burden had a significant effect on long-term mortality in the non-frail male population. (B) Shows that the level of anticholinergic burden had no significant effect on long term mortality in the frail male population. (C) Shows that the level of anticholinergic drugs had a significant effect on long-term mortality in the non-frail women population. (D) Shows that the level of anticholinergic drugs had a significant effect on long-term mortality in the frail women population.

tool (*Gasser et al., 2020*). In our study, the prevalence of frailty in the outpatient setting of a tertiary referral hospital was 44%. Although we used the CHS phenotypic criteria to identify frailty, our result is similar to that of the study by *Gasser et al. (2020)*. In addition, *Chen et al. (2010)* indicated that the prevalence of frailty, prefrailty, and non-frailty in community-dwelling Taiwanese were 4.9%, 40.0%, and 55.1%. To our knowledge, this is the first study to report the prevalence of frailty in the older Taiwanese patients in outpatient clinics. The results of our study provide insight into sex-specific mortality risks with regard to frailty, polypharmacy, and anticholinergic bruden to enable the development of potential strategies for managing frailty in this vulnerable population.

Since frailty will cause a variety of adverse medical effects, it is necessary to correct the risk factors of frailty and reduce the associated mortality risk. Digital Health Intervention involves the enhancement of diverse health conditions, encompassing frailty assessment and intervention. While numerous studies have employed digital health interventions among frail older populations, the advantages of their utilization still necessitate more standardized approaches for frailty assessment and intervention, alongside meticulously

designed randomized controlled trials (*Linn et al., 2021*). Based on existing evidence, it appears that telehealth-based medication reviews could offer a viable framework for providing such services, with the potential to yield cost savings and enhance patient care. Nonetheless, the current level of evidence might not be substantial enough to confidently guide practice and policy decisions regarding telehealth-facilitated medication reviews (*Shafiee Hanjani et al., 2020*).

Many studies have shown that the prevalence of frailty is higher in females than in males, even in the same age group. However, the mortality rate of older men is higher than that of older women (*Collard et al., 2012*; *Gordon et al., 2017*), which is known as the "sex-frailty paradox" (*Gordon et al., 2020*; *Gordon et al., 2017*). Furthermore, our findings support the abovementioned paradox. Sex differences in genetic, biological, or psychosocial factors as well as in physiological reserve, burden of disease, and disability might contribute to this paradox (*Gordon et al., 2020*; *Gordon et al., 2017*). Our data suggested that a potentially unrecognized factor –polypharmacy, especially drugs with anticholinergic effect—might contribute differentially to mortality in the male and female groups. However, *Pietrantonio et al. (2023)* found that the number of drugs used did not show a significant impact on mortality in male and female patients affected by both COVID-19 and cardiovascular diseases. Further studies are needed to confirm our findings.

In addition to frailty, the geriatric syndrome is associated with mortality in the older population (*Ates Bulut, Soysal & Isik, 2018*; *Huang et al., 2017*; *Kane et al., 2012*). A longitudinal cohort study conducted at the outpatient geriatric clinic of Amsterdam University Medical Center classified the common geriatric syndrome into five domains (somatic, mental, nutritional, physical, and social) and the result showed that the geriatric syndrome was cumulatively associated with mortality (*Rhodius-Meester et al., 2021*). These results raise an intriguing possibility that the interaction between the geriatric syndrome and frailty may be partly responsible for the sex-frailty paradox. Our study found that polypharmacy is an important determinant of mortality and confers different effects between both sexes as well as between groups with and without frailty. We speculated that polypharmacy may be one of the causes of the sex-frailty paradox. Polypharmacy is highly prevalent in the older population and is as high as 46.6% of the geriatric population. The higher the number of drugs that are used, the higher the risk of hospitalization and death (*Chang et al., 2020*). A systemic review of 25 observational studies demonstrated a significant association between polypharmacy and frailty (*Gutierrez-Valencia et al., 2018*). *Gasser et al. (2020)* demonstrated that the prevalence of polypharmacy in the frail group was significantly higher than that in the non-frail and pre-frail groups. Moreover, *Bonaga et al. (2018)* revealed that polypharmacy and frailty status are interrelated, and their interaction determines the frequency of health-related adverse events. Polypharmacy and frailty were associated with a significantly higher incidence of incident disability, hospitalization, and mortality, compared to non-frail individuals without polypharmacy (*Bonaga et al., 2018*). The findings from the European dataset, specifically the Survey of Health, Ageing, and Retirement in Europe cohort, revealed that polypharmacy has a greater impact on long-term mortality in the non-frail group when compared to the frail group (*Midao et al., 2021*). Moreover, this finding is supported by the results of another

population-based cohort study that was conducted by using Taiwan's National Health Insurance Reimbursement Database (*Chen et al., 2021*). Within the same polypharmacy category, the grade of frailty status was associated with an adverse effect on mortality. Interestingly, the dose–response association between polypharmacy and mortality was only observed in the fit and mild-frail participants (*Chen et al., 2021*). Our findings that polypharmacy modulates the 5-year mortality risk in the non-frail, but not in the frail group supports the results reported from the abovementioned studies. Nevertheless, the presence of a sex difference in the interaction between polypharmacy and frailty was previously unknown. We found that polypharmacy might contribute to the long-term survival outcome only in the male and non-frail older adult group. Our data support the assumption that avoiding polypharmacy and inappropriate medication might improve the risk of long-term mortality, especially in male, non-frail groups.

Apart from the challenge of polypharmacy among the elderly, the adverse effects of medications, particularly the anticholinergic effects, pose significant concerns for clinicians. Epidemiological surveys highlight that half of the older adult population is prescribed at least one medication with anticholinergic properties. This prevalence can be attributed to their prescription for various therapeutic indications, including the management of urinary incontinence or sleep disorders. Individuals aged 65 years and older are more vulnerable to anticholinergic effects due to a range of physiological changes associated with aging. These changes include reduced renal and hepatic function, affecting drug clearance, alterations in body weight distribution, and increased permeability of the blood–brain barrier (*Laatikainen et al., 2016*; *Mangoni & Jackson, 2004*; *Villalba-Moreno et al., 2016*). A comprehensive review by *Mehdizadeh et al. (2021)* highlights that the exposure to anticholinergic medications among older individuals with frailty has an impact on grip strength and gait speed. However, its influence on mortality is primarily confined to those in a state of pre-frailty and frailty, particularly evident in association with antipsychotic drugs (*Mehdizadeh et al., 2021*). Our study investigates the influence of anticholinergic effects on frail populations, incorporating a subgroup analysis based on gender. Our findings reveal a notably significant anticholinergic effect among frail women. It is worth noting that this study stands as a unique exploration into the ramifications of anticholinergic agents on long-term mortality within frail populations.

There are some limitations of our study. First, our study population was recruited from the geriatric clinics of a single hospital in Taiwan. Our result might not be generalizable to the general older outpatient and patients of non-Han ethnic group. Second, we focus solely on analyzing drugs with anticholinergic effects, and there is no in-depth analysis conducted for drugs in other categories.. Therefore, we could not confirm which drug classes might have a deleterious impact on long-term mortality. Third, this study did not analyze the cause of death. Thus, the presence of a causal relationship between polypharmacy, anticholinergic burden, frailty, and death needs to be ascertained. However, this study provided valuable information that polypharmacy and anticholinergic burden might contribute significantly to mortality in the non-frail, male older adult group. Further prospective studies are needed to determine whether a reduction of polypharmacy offers long-term survival benefits to geriatric patients, with or without frailty.

## CONCLUSION

This study demonstrated sex differences in the additive effect of polypharmacy and greater anticholinergic burden in the 5-year mortality risk in the non-frail group. To improve long-term survival, physicians should identify and tackle polypharmacy and anticholinergic burden stringently, especially in older non-frail adults.

## ACKNOWLEDGEMENTS

We thank the Biostatistics Task Force of Taichung Veterans General Hospital for their assistance with the statistical analysis of the research data that are discussed in this paper. Thank you to the Division of Clinical Information, Center for Quality Management of Taichung Veterans General Hospital for assisting us in extracting participants' medication information from the medical records database.

### Funding

The authors received no funding for this work.

### Competing Interests

The authors declare there are no competing interests.

### Author Contributions

- Yin Yi Chou conceived and designed the experiments, performed the experiments, analyzed the data, prepared figures and/or tables, authored or reviewed drafts of the article, and approved the final draft.
- Yu Shan Lee conceived and designed the experiments, authored or reviewed drafts of the article, and approved the final draft.
- Chu Sheng Lin performed the experiments, authored or reviewed drafts of the article, and approved the final draft.
- Jun Peng Chen performed the experiments, prepared figures and/or tables, and approved the final draft.
- Fu-Hsuan Kuo analyzed the data, authored or reviewed drafts of the article, and approved the final draft.
- Cheng-Fu Lin analyzed the data, authored or reviewed drafts of the article, and approved the final draft.
- Yi-Ming Chen conceived and designed the experiments, performed the experiments, analyzed the data, prepared figures and/or tables, authored or reviewed drafts of the article, and approved the final draft.
- Shih-Yi Lin conceived and designed the experiments, performed the experiments, authored or reviewed drafts of the article, and approved the final draft.

## Human Ethics

The following information was supplied relating to ethical approvals (i.e., approving body and any reference numbers):

the Ethics Committee of Clinical Research, Taichung Veterans General Hospital

## Data Availability

The raw measurements are available in the Supplementary Files.

## Supplemental Information

Supplemental information for this article can be found online at http://dx.doi.org/10.7717/peerj.16262#supplemental-information.

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
