# Peer review of "Gender-specific linkages: frailty, polypharmacy, anti-cholinergic burden, and 5-year mortality risk—a real-world analysis"

_PeerJ, doi:10.7717/peerj.16262_

## Round 0.1 · original submission · Major Revisions

Dear Dr. Lin,

Your manuscript entitled “Gender-specific association of frailty, polypharmacy, and the five-year mortality risk: a real-world retrospective study" which you submitted to PeerJ, has been reviewed by the editor and 3 external reviewers.

I regret to inform you that the reviewers have raised some significant concerns that need to be addressed before the manuscript can be considered further. However, since reviewers do find some merit in the paper, I would be willing to reconsider if you wish to undertake major revisions and resubmit.

If you decide to resubmit the revised version, please summarize all the improvements made in the new version and give answers to all critical points raised in the reviewers’ report in an accompanying letter. Copy and paste each and every reviewer's comment above your response.

Please note that resubmitting your manuscript does not guarantee eventual acceptance. Since the requested changes are major, the revised manuscript will undergo a second round of review by the same reviewers. I must emphasize that the acceptability of the revision will depend upon the resolution of the points raised by the reviewers.

Sincerely yours,
Stefano Menini

Reviewer 1 ·

Basic reporting

dear Authors, thank you for your work. frailty is one of the biggest problems of our era. I checked your bibliography: some articles are published before 2010 and are, in my opinion, too old. in 2023
I find some recent articles: https://pubmed.ncbi.nlm.nih.gov/37174229/
2) what about telehealth and frailty?

Experimental design

this section is ok

Validity of the findings

this section is ok

Additional comments

no

Reviewer 2 ·

Basic reporting

Thank you for the opportunity to read the manuscript “ Gender-specific association of frailty, polypharmacy, and the five-year mortality risk: a real-world retrospective study” first.
It is clearly written, unambiguous and fits into the field of geriatric syndromes. The conclusions have been carefully considered and argued, the tables and figures are instructive.

Experimental design

It fits well in the scope of the journal, include a well defined research question, the methods are adequate presented.

Validity of the findings

Definition of polypharmacy is insufficient in this context of the study. Not the pure number of medication but inappropriate medication, over-or undertreatment and interaction of medication with ongoing diseases cause morbidity and mortality. It is of major concern for the predictive power of all following analysis.

Additional comments

1. Line 67: correct the reference
2. Line 126: Definition of polypharmacy is insufficient in this context of the study.
3. Line 239: comma mistake
4. Line 252: as forth limitation - the lack of qualitative analysis of ongoing medication, e.g. medication appropriateness index or any explicit/implicit tool for diagnosing polypharmacy, is missing.

Reviewer 3 ·

Basic reporting

Kindly add the author name in the 2021 reference in line 67.
Line 74: Consider changing 'increases with higher age and female sex' to 'increases with higher age and in females'
Line 75- consider removing the word 'the'

Experimental design

The research is original, and adds to the existing literature about polypharmacy, frailty and mortality. The research question is define, and the investigation methods used are sound. The methods were described well.

Validity of the findings

The results are meaningful and there is sufficient rationale behind it. The data is provide dis robust and statistically sound. The conclusions are well stated.

Additional comments

Overall an interesting paper. The manuscript adds to the large body of evidence pertaining the prevalence of polypharmacy and frailty. There is not much work done in regards to a 5-year all cause mortality and sex-specific mortality risks, so that is the novelty of this paper.

---

## Round 0.2 · accepted · Accept

Dear Dr. Lin,

Thank you for submitting a revised version of your manuscript. I am pleased to inform you that your manuscript is accepted for publication in PeerJ in its current form.

I thank all reviewers for their effort in improving the manuscript and the authors for their cooperation throughout the review process.

Sincerely yours,
Stefano Menini

Reviewer 1 ·

Basic reporting

dear, thank you for your paper. the sections are very well builted. you shouldn't use articles older than 5 years. data are suitable and logic.

Experimental design

it's ok

Validity of the findings

it's ok

Reviewer 2 ·

Basic reporting

It is clearly written, unambiguous and fits into the field of geriatric syndromes.

Experimental design

It now fits in the scope of the journal.

Validity of the findings

The conclusions have been carefully considered and argued, the tables and figures are instructive.

Additional comments

Thank you for the opportunity to read the revised manuscript on “Gender-Specific Linkages: Association of Frailty, Polypharmacy, Anti-Cholinergic Burden, and the Five5-year Mortality Risk – : A Real-World Analysis Retrospective Study”. All suggestions made in my review were carefully mentioned in the point to point reply and subsequently adapted in the revised manuscript. In my point of view it is well prepared for publishing.